# Continued Dispensing: what medications do patients believe should be available?

Salem Hasn Abukres, Kreshnik Hoti and Jeffery David Hughes

School of Pharmacy and Curtin Health and Innovation Research Institute, Curtin University, Perth, Western Australia, Australia

## ABSTRACT

**Background**. Continued Dispensing (CD) is a new medication supply method for certain medications in Australia. It aims to prevent treatment interruption as a result of patients' inability to obtain a new valid prescription. The only currently eligible patients for this service are statin and/or oral contraceptives users who have been using these medications for 6 months or more, have not utilized the CD method during the last 12 months, and cannot obtain an immediate appointment with the prescriber in order to get a new prescription. This study aimed to investigate patients' attitudes towards potential extension and expansion of this medication supply method.
**Methods**. A randomly selected 301 users of these medications from all Australian States were recruited using Computer Assisted Telephone Interview (CATI).
**Result.** The response rate was 79%. The majority of the participants (73.3%) did not agree with current restriction on CD utilization frequency. They also supported, to varying degrees, inclusion of all the proposed medications (support ranged from 44.2–78.4%). In this regard, participants who suffered from a specific disease did not differ significantly from those without the disease except in case of patients with depression ($p = 0.001$).
**Conclusions.** Participants of this study strongly supported both CD extension and expansion. A future critical review of the current version of CD is highly recommended in order to enhance CD capability to achieve its goals.

## INTRODUCTION

Recently in Australia a new method of medications supply, Continued Dispensing (CD), has been implemented to provide patients with a more convenient way to obtain their medications. Patients who run out of statins or oral contraceptives (OCs) are no longer required to present a valid prescription to request these medications under the following conditions: they are unable to obtain an immediate appointment with their doctor, they have been using the medication for more than 6 months, and have not utilized the CD method during the last 12 months (*5th Community Pharmacy Agreement, 2013*). This system was originally proposed to minimize the risk of patients running out of their medication between doctors' visits (*Bessell et al., 2005*).

Corresponding author
Salem Hasn Abukres,
salamhasn.abukres@postgrad.
curtin.edu.au

Medications in Australia are available as: Prescription only medications (Schedule 4, and Schedule 8 for controlled drugs), and non-prescription medications, which include: Pharmacist Only (Schedule 3) which can be provided by a pharmacist, Pharmacy Only (Schedule 2) which can be provided by other pharmacy staff under pharmacists' supervision, and other non-scheduled over the counter medications (OTC) which are available for general sale. Statins and OCs (except for the emergency contraceptive pill) are S4 medications, (i.e., a prescription is required for dispensing). However, pharmacists are authorized to dispense these and other types of prescription only medications without a prescription under certain circumstances; for example, as an emergency supply if the patient is a regular user of the requested medication, has no access to the medication and lacks a prescription for any valid reason, e.g., traveling or out of date prescription. In such circumstances, the pharmacist may offer a three day supply (*Bessell et al., 2005*). Another method to dispense statins or OCs without a prescription is according to the recently introduced CD. According to this method pharmacists can dispense one additional supply (i.e., one standard pack) of the medicine, which is generally enough for one month. Previous literature data suggested that CD eligible patients strongly supported this method (*Abukres, Hoti & Hughes, 2014*). Furthermore, the Owing Prescription system is used to continue the supply of medications if obtaining an appointment with the prescriber is not practical. However, this system requires the pharmacist to contact the prescriber before dispensing for approval and the prescriber must provide a written prescription within seven days (*Bessell et al., 2005*; *Hoti, Hughes & Sunderland, 2012*).

The current CD method may provide limited benefit to chronic disease sufferers as they are often on multiple medications (*Hughes, 2005*). Consequently, they may present to a pharmacy requesting a statin or an OC, as well as other medications which are currently not eligible for supply under the CD model. In these situations, conducting CD may confuse the patient who can, for instance, obtain their statin without a prescription but cannot obtain their antihypertensive medication. In this case, CD may not be an appropriate service to offer.

The above suggests that the list of CD eligible medications may need to be expanded to cover a wider range of common diseases, as is the case with pharmacist supplementary prescribing models in other countries. Pharmacist supplementary prescribing is a partnership between doctors and pharmacists where doctors retain their diagnostic role (*Hoti, Hughes & Sunderland, 2011a*). Patients who experienced supplementary prescribing have shown their support for pharmacists to prescribe a variety of medications such as, but not limited to, medications to treat diabetes, epilepsy, cancer, cardiovascular, respiratory, renal, skin, gastrointestinal, thyroid and blood coagulation diseases (*Lloyd, Parsons & Hughes, 2010*). Furthermore, pharmacist interventions with treatment of chronic diseases have been proven to be effective (*Fikri-Benbrahim et al., 2013*; *George et al., 2010*; *Smith et al., 2010*). Therefore, applying more responsibility to pharmacists through an expanded version of CD may assist in achieving CD's goals; i.e., a more convenient way for patients to obtain their medication in a timely manner, prevent treatment interruptions, utilize pharmacists' skills and decrease overload on doctors. The Australian Medical Association (AMA)

has described the current (limited) CD as unsafe and inappropriate (*Australian Medical Association., 2012*). This was in contrast with results of a survey of statin and (OC) users, where the majority of the respondents did not perceive CD would pose any risks (*Abukres, Hoti & Hughes, 2014*). They also trusted their pharmacists would conduct CD only when it was safe to do so and that their pharmacist would refer them to their doctor when needed. Furthermore, they thought pharmacists are more easily accessible than doctors; that CD would save their and their doctor' time, and that it would help them to not miss any doses of their medications (*Abukres, Hoti & Hughes, 2014*). It is worth mentioning that patients surveyed had no personal experience with CD as the study was conducted before the actual implementation of CD in Australia, so the results represent participates' perceptions rather than their actual experience. Moreover, patients are not necessarily qualified to identify precisely when it's safe or not to obtain a medication without a prescription.

Another limitation of the CD is its restriction to be conducted only once in any 12 month period. This timeframe has been proposed to prevent patients avoiding doctors' visits (*Bessell et al., 2005*). It may lead, however, to treatment interruptions in two ways: if the additional supply is not enough until the next available appointment with the doctor (*Grudzen et al., 2011*; *Viberg et al., 2013*), and/or if the patient runs out of valid prescriptions for their medication more than once in a 12 month period. This may occur as a result of 6 months coverage of chronic medication prescriptions (*Britt et al., 2011*).

In this study we sought to explore patients' attitudes towards expansion of CD to include a broader range of medications and hence increasing the access to the service. The study was conducted before the actual implementation of CD on September 1, 2013. In doing so, we were able to assess respondents' attitudes before they had experienced the service, and hence without any bias of personal experience.

## METHODS

### Study design and sampling

A more detailed methodology of this study has been reported elsewhere (*Abukres, Hoti & Hughes, 2014*). Computer-assisted telephone interviewing (CATI) was used. A stratified sample was used. The target sample size was at least 150 statin and 150 OC users, which allowed for a prevalence estimation of ±5%. A telephone number list of a total of 25,000 records was randomly generated based on a broad breakdown of the Australian population by state and territory as outlined in the June 2013 Australian Bureau of Statistics data (i.e., New South Wales 32%, Victoria 25%, Queensland 20 %, South Australia 7 %, Western Australia 11%, Australian Capital Territory 2%, Tasmania 2% and Northern Territory 1%) (*Australian Bureau of Statistics, 2013*). The eligible candidates were users of statins and/or oral contraceptives who were ≥18 years old, spoke English, and used landline phones. Participants who used both medications were interviewed as OC users. Participants were considered consented if they agreed to the question: Would you like to participate? They were told that the interview would take up 15 minutes and they could withdraw at any time. They were called once and if the call timing was not convenient for the interviewee they were called again when it was convenient. Participation was voluntarily

**Peer**J

and no incentives were used. Ethics approval for the study was obtained from The Human Research Ethics Committee of Curtin University (Approval number: PH-06-13) (*Abukres, Hoti & Hughes, 2014*).

## Questionnaire design

The participants were interviewed using a questionnaire consisting of 38 closed ended questions, with the option 'other: please specify' for some questions. The responses were entered into a database verbatim. The questionnaire was developed through a literature review, and experience from a previous study (*Hoti, Hughes & Sunderland, 2011b*). The main focus of the questionnaire was to identify areas of CD extension and expansion, particularly what other medications may potentially be included into the CD system. The challenge was to carefully select potential medication classes and medical terms that were easily understandable by the general population. The study tool was assessed for face and content validity by staff members within the pharmacy practice group at Curtin University and by the telemarketing company group CDM Direct Communication Services. The final questionnaire (abridged version Appendix S1) was used by the CDM's trained staff (thus minimizing bias) to collect data during July 2013 (*Abukres, Hoti & Hughes, 2014*).

## Statistical analysis

The Statistical Package for the Social Sciences (SPSS$^{TM}$) version 22 was used for statistical analysis. In this regard, frequency distribution analysis and Chi square test were employed to assess associations between variables. Answers were collected on a 6 point Likert scale (where 1 = Strongly disagree, 2 = Disagree, 3 = Neutral, 4 = Agree, 5 = Strongly agree, and 6 = Prefer not to disclose). For the analysis purposes, the scale was trichotomized as follows: Agreed, which included options 4 and 5, and Disagreed, which included options 1 and 2, and Neutral which included option 3.

## RESULTS

### Sample demographic characteristics and response rate

Some of these results have previously reported elsewhere (*Abukres, Hoti & Hughes, 2014*). There were 10,479 attempts to make phone calls. However, there were a large number of failed calls (7,019) due to various reasons, including: number disconnected, answer phones, answering machines, no answer or call busy. This resulted in 3,460 successfully contacted individuals. Among these 2,443 individuals were screened out because of ineligibility ($n = 2,146$ who were either under 18 years or not taking a statin or OC, and 297 respondents were deemed unable of completing the survey because of language or hearing difficulties). In addition, there were 716 outright refusals, these were the most problematic because they refused to participate at an early stage before it was clear if they were eligible or not. Since the outright refusals were likely to have a mixed eligibility, the 716 outright refusals were considered to have the same proportion of eligible individuals as the screened sample (i.e., 11% which was calculated by dividing 301 respondents by 2,744). This resulted in an estimated 380 total eligible candidates, giving a response rate of 79%
(301 respondents of the total 380 eligible individuals). However, if all the outright refusals were eligible, which is highly improbable, the response rate could have been as low as 30%. Irrespective of the response rate the targeted sample size was successfully obtained.

The respondents were made up of 151 statin and 150 OC users, with the sample consisting of a lower number of male participants compared to females (20% vs 80%, respectively). Their ages were distributed almost equally into those 60 years or younger and those older than 60 years. The participants were recruited from all the Australian states and territories; however, the state of New South Wales donated almost half of the participants. Only 16% of the participants were from rural areas (*Abukres, Hoti & Hughes, 2014*).

## Participants' attitudes towards expansion of CD

### CD extension: increased access to CD

Participants were asked how many times they thought CD should be allowed within a 12 month period. The majority of participants (73.3%; $n = 222$) disagreed with the current CD limitation, and selected more than one CD within a 12 month period. Among those who wanted more CD; 16.3% selected 'twice a year,' 5.9% selected 'three times a year,' and 51.1% of the participants selected 'any time my repeats run out and I am not able to get an appointment with my doctor.'

### CD expansion: addition of more medications to the current CD list

Participants were asked about their thoughts (i.e., agreement levels) on expanding the current list of CD eligible medications. Table 1 shows the proportion of participants who agreed with inclusion of medications for specific diseases/disorders.

Respondents' support to include particular medications was affected by the condition to be treated. For example, 78.4% ($n = 236$) of the participants agreed to the inclusion of asthma medications; however, only 44.2% ($n = 133$) agreed to the inclusion of

**Table 1 Respondents' preferences for medications to be covered under CD.**

| Disease/disorder/medication | Neutral/did not disclose $n$ (%) | Disagreed $n$ (%) | Agreed $n$ (%) |
|---|---|---|---|
| Asthma | 34 (11.3) | 31 (10.3) | 236 (78.4) |
| Arthritis | 39 (13.0) | 35 (11.6) | 227 (75.4) |
| Chronic skin disorders | 40 (13.3) | 39 (13.0) | 222 (73.8) |
| Indigestion | 42 (14.0) | 40 (13.3) | 219 (72.8) |
| Hypertension | 40 (13.3) | 48 (15.9) | 213 (70.8) |
| Diabetes | 62 (20.6) | 37 (12.3) | 202 (67.1) |
| Chronic bronchitis | 58 (19.3) | 55 (18.3) | 188 (62.5) |
| Emphysema | 67 (22.3) | 59 (19.6) | 175 (58.1) |
| Chronic pain | 54 (17.9) | 86 (28.6) | 161(53.5) |
| Blood clotting | 62 (20.6) | 85 (28.2) | 154 (51.2) |
| Thyroid | 84 (27.9) | 67 (22.3) | 150 (49.8) |
| Glaucoma | 80 (26.6) | 71 (23.6) | 150 (49.8) |
| Anxiety | 58 (19.3) | 98 (32.6) | 145 (48.2) |
| Depression | 55 (18.3) | 113 (37.5) | 133 (44.2) |

antidepressants. The participants' support for the inclusion of different medications can be divided into three levels based on level of agreement. Level 1: included medications to treat asthma, arthritis, chronic skin problems, indigestion, hypertension, diabetes (oral hypoglycaemics) and chronic bronchitis, where over 60% of the participants supported their inclusion within the CD provision. Level 2: included emphysema medications, chronic pain medications, and anticoagulants where more than 50% (but less than 60%) of the participants agreed to their inclusion, and Level 3: included medications for thyroid disorders, glaucoma, anxiety and depression, which were supported by less than 50% (Table 1).

## Views of other disease suffers

More than one third of participants suffered from other chronic diseases (38.8%; $n = 116$). The most prevalent co-morbidities were hypertension, type 2 diabetes mellitus, arthritis, depression, asthma, ingestion, and thromboembolic disorders requiring anticoagulation. Table 2 compares the views of participants with these particular diseases with those without. Generally, in all diseases except type 2 diabetes and indigestion, the proportion of disease suffers who agreed with their medication's inclusion into the CD provisions was higher than the proportion of the total study cohort. However, the only statistically significant difference was between participants supporting inclusion of antidepressants, where participants with depression supported inclusion of these medications more than participants without this disorder (92.8 vs 41.8%, $p = 0.001$).

## DISCUSSION

To the best of our knowledge, this is the first study to explore the views of statin and OC users in regards to potential extension and expansion of the CD system in Australia. More specifically, their support to increase the maximum number of times that CD can be utilized within a 12 month period (i.e., CD extension) and expansion of the range of medications allowed to be dispensed under CD (i.e., CD expansion). Regarding CD extension, the majority of the participants disagreed with the current restriction of CD to once in every a 12 month period, and preferred the option of using it more frequently. Interestingly, more than half of the participants wanted CD to be available until it was

**Table 2** Influence of experience with the disease on respondents' attitudes to the inclusion of particular medications in CD.

| Disease | Agreed participants without the disease $n$ (%) | Agreed participants with the disease $n$ (%) | $P$ value |
|---|---|---|---|
| Hypertension | 180 (68.7) | 33 (84.6) | 0.06 |
| Diabetes mellitus | 186 (67.4) | 16 (64.0) | 0.91 |
| Arthritis | 211 (75.1) | 16 (80.0) | 0.81 |
| Depression | 120 (41.8) | 13 (92.9) | 0.001 |
| Asthma | 225 (77.9) | 11 (91.7) | 0.23 |
| Indigestion | 214 (72.8) | 5 (71.4) | 0.61 |
| Blood clotting | 149 (50.5) | 5 (83.3) | 0.35 |

possible for them to see their doctor. This may indicate that patients required more flexibility to avoid unnecessary treatment interruption if, for any reason, an appointment with their doctor could not be achieved. Previous studies have reported that patients have difficulty in seeing their regular doctor without a prescheduled appointment (*Gallagher et al., 2001*; *Garth et al., 2014*). Furthermore, it has been reported that patients often do not organize appointments in advance or failed to attend appointments (*Minty & Anderson, 2004*).

On the second question regarding expansion of the medications available through CD, participants generally supported inclusion of more medication classes. However, this support was influenced by the use of those medications. In this regard, the lowest level of support was for medications for the treatment of depression and the highest support was for asthma medications. This profound support for the inclusion of medications to treat a broad range of diseases/disorders may be related to patients' confidence in their self-management and the ability to judge the severity of these diseases. Additionally, this may be related to their confidence that pharmacists can provide monitoring for diseases such as diabetes and hypertension. In a previous study (*Wakefield et al., 2000*), patients provided reasons for preferring to buy short acting beta agonists (SABAs) without a prescription or with repeats of a previously issued prescription rather than visiting their doctor and obtaining a new prescription after a clinical examination. These reasons included their perception of the worthlessness of visiting their doctor just to obtain a new prescription, their perceptions of medication not requiring such visits and their long experience with the disease, making them feel that they were able to manage and control asthma without the need to see a doctor. This is despite evidence by Braido that "self-reported symptoms poorly correlate with pulmonary function measures" (*Braido, 2013*). Another study reported that obtaining SABAs without a prescription did not lead to poorer asthma control; instead, it supported the claim that OTC availability of these medications benefits patients with asthma (*Douglass et al., 2012*). Moreover, the availability of some medications to treat asthma such as (SABAs) as a Pharmacist Only medications in Australia, that do not require a prescription, may have increased participants' confidence to obtain more asthma medications without a valid prescription. On the other hand, other studies reported that OTC asthma medicines have resulted in under-treatment and less consultation with doctors. Furthermore, assessment and counselling provided by pharmacists or other pharmacy staff has been reported to be less than optimal (*Schneider et al., 2009*). However, this inadequate counseling may have resulted from unwillingness of patients with long term chronic diseases to discuss with healthcare professionals what they believed they already know. This controversy about effectiveness and benefit of dispensing asthma medications without a doctors' review raises the need to ensure that optimal patient outcomes are being achieved through appropriate monitoring. This suggests that down scheduling of Prescription Only Medication to Pharmacist Only Medication provides better access to those medications (*Gauld et al., 2014*). However, appropriate patient supervision is essential, as is referral to the doctor whenever deemed necessary (*Abukres, Hoti & Hughes, 2014*).

Disease sufferers were more likely to support inclusion of their medications into CD with the exception of patients with diabetes mellitus and indigestion. This is probably because low sample size of patients suffering from ingestion. Interestingly, more than 92% of patients with depression supported inclusion of antidepressants in CD, even though the overall support for the inclusion of medication for depression was the lowest. The exact reason for the difference in support for the inclusion of antidepressants is unclear. However, it may reflect a poorer level of mental health literacy amongst the general population without depression, in which diseases like depression still have a social stigma. The lack of support for the availability of anxiolytics may also be explained in the same way, although the potential for abuse of these medications may also be another explanation. On the other side, the fact that most antidepressants are labelled with warnings "do not stop abruptly" may explain why patients with depression were more supportive than the overall participants. In addition, patients on long-term treatment for depression may not see the need for another visit to the doctor, especially if they do not perceive that they receive any new information during their routine appointments (*Gask et al., 2003*). The later may apply to chronic diseases in general, where patients after several years on the same medication may accept that nothing will be change and see their doctor's appointments as adding little value to their management.

Participants' support to include additional medications under the CD provisions is consistent with practices in Canada, where pharmacists in some provinces are permitted to undertake short-term dispensing to allow patients to avoid interruption of their continuing therapies (*Law et al., 2012*). It is also consistent with the overall trend to expand pharmacists' roles through rescheduling more prescription only medicines to non-prescription status that requires additional pharmacist's intervention (i.e., Pharmacist Only Medications). This includes medications to treat asthma, hypertension and hyperlipidemia (*American Society of Health-System Pharmacists, 2012*; *Gabay, 2013*; *Gauld et al., 2014*). Conversely, doctors have expressed their concerns about safety and appropriateness of CD (*Daniels, 2012*), as well as any further reclassifying of prescription only to non-prescription status (*Calabretto, 2012*).

Limitations of this study have been reported in detail previously (*Abukres, Hoti & Hughes, 2014*). These include the respondents' distribution, with almost half of the participants being from one state (i.e., New South Wales); other factors included the exclusion of those who were under 18 years old, did not speak English, did not use landline phones or were not available at the time of calling. In regard to this section of the study, participants' support to include more medications in the CD model was not based solely on their personal experience, as no participant had all the listed diseases/disorders. In addition, the number of participants who suffered from other diseases were low; therefore their support to include these medications may not be generalizable. However, participants' responses may reflect, amongst other things, their general awareness of the disease/disorder, the experience of a friend or relative and/or being a health care professional. The fact that the participants were all either statin or OC users, may bias the results as they may have different views from the general public or patients with the specific

diseases, which may limit generalizability of the results of the current study. However, there are factors that contribute towards the study strengths, such as being the first about patients' views on the current CD system it may provide insights into how the system may be extended and expanded in the future.

Future research should explore specific diseases in relation to CD, including clinical and economic implications. Further, it is important that research undertaken to assess whether patients' expressed desire to expand and extend CD is in their best health and economic interests. Given the limitations of the current CD method, other medication supply models to patients in cases where there is lack of valid prescription should also be explored.

## CONCLUSIONS

Current restrictions on CD may limit its capacity to serve its goals, as suggested by this study with participants highly supporting a more flexible and broader CD system. The currently eligible utilizers of the CD system seem to prefer inclusion of additional medications and more opportunity to use CD at any time they cannot see their doctor. These findings suggest that ongoing review of CD is essential and changes which do not compromise patient safety or allow the abuse of CD would be welcomed by patients.

### Funding
The authors declare there was no funding for this work.

### Competing Interests
The authors declare there are no competing interests.

### Author Contributions
- Salem Hasn Abukres, Kreshnik Hoti and Jeffery David Hughes conceived and designed the experiments, analyzed the data, contributed reagents/materials/analysis tools, wrote the paper, prepared figures and/or tables, reviewed drafts of the paper.

### Human Ethics
The following information was supplied relating to ethical approvals (i.e., approving body and any reference numbers):

Ethics approval for the study was obtained from The Human Research Ethics Committee of Curtin University (Approval number: PH-06-13).

### Data Deposition
The following information was supplied regarding the deposition of related data:

Raw data for this article is available upon request by contacting Salem Abukres (salamhasn.abukres@postgrad.curtin.edu.au).

### Supplemental Information
Supplemental information for this article can be found online at http://dx.doi.org/10.7717/peerj.924#supplemental-information.

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
