# Peer review of "Continued Dispensing: what medications do patients believe should be available?"

_PeerJ, doi:10.7717/peerj.924_

## Round 0.1 · original submission · Major Revisions

Please note, in particular Reviewer 2's comments in relation to the response rate before deciding to re-submit the work.

I would also like to see a much more refined Methods Section of the paper and more explicit discussion surrounding limitations.

·

Basic reporting

This is generally a well written manuscript that outlines some results from a telephone survey regarding continued dispensing. Whilst the authors note that the methods have been previously published, when I reviewed the original published manuscript the questionnaire is not available and I feel that the pertinent original questions would help the reader interpret some of the findings. Further in the original manuscript already published it says the questionnaire was validated by staff members, but not the exact method of validation which should be added.
There are some minor wording change suggestions:
1. on page 2 where the S2 and S3 schedules are explained, OTC is referred to as non scheduled. AS all of these are technically OTC i think some minor edits could occur here.
2. On page 4 part A. I do not understand the subtle difference between the 2 results presented, 73.3% and 51.1%. it looks like they all wanted more than 1 CD available in a 12 month period. Is the 51% referring to the reason? (ie 51% believed it would allow them to see a doctor?) It is nor clear.
3. Page 5 in the first paragraph of discussion reword "every a 12 month"
4. Page 5 regarding asthma it is recommended to call the patients with asthma rather than asthma patients
5. On page 6 insert "of" between the words inclusion and antidepressants and insert the word "in" between disease and general

Experimental design

As above the validation of the survey and inclusion of the pertinent questions is required.

Other issues to understand regarding the design is the rational for coding 3 (neutral) as did not agree? As the results are presented this starts to make sense, in fact all you have reported are those with a 5 or 4 but the way it reads in the methods was initially confusing. I wonder if it would be interesting to report those who disagreed? so were many people neutral towards antidepressants or did they disagree for example?

Validity of the findings

The research shows that patients agree that other classes of medicines should be added and that this was not necessarily influenced strongly by their own disease state.

the samples however for these secondary analyses are low so it may be difficult to draw strong conclusions about these findings.

Additional comments

nil

Reviewer 2 ·

Basic reporting

The introduction would benefit from being a little less Australian-centric. Does continuation dispensing occur in other countries, and if so, how (briefly)? Supplementary pharmacist prescribing is a little different – is there any other model quite like Australia? I think the S3, S2, S4 and S8 adds unnecessary complexity given that this is not especially about scheduling in Australia as such. For example, say Statins and oral contraceptives (except for the emergency contraceptive pill) are prescription medications. I would not say pharmacy only medicines “can be prescribed by other pharmacy staff”. I doubt that most readers would consider a pharmacy only medicine supply by a non-health professional in Australia “prescribing”. One definition of prescribing is “to order a medicine or other treatment” which makes pharmacist-only supply not always prescribing either.
The following sentence (which is very long) would benefit from being broken into two components
Although the Australian Medical Association (AMA),(Australian Medical Association 2012) has described the current (limited) CD as unsafe and inappropriate, this was not supported by the results of a survey of statin and oral contraceptive users, where the majority of the respondents did not perceive CD would pose any risks.(Abukres et al. 2014)
The fact that a survey of consumers did not think there would be any risks is in contrast to the earlier statement rather than not supporting it given a difference in knowledge and understanding of potential medical risks by the public versus health professionals.

Experimental design

The design is rather too brief – do not assume most readers will look elsewhere. Study dates should be in the methods design not the introduction. Where the respondents were drawn from should be in the study design, and whether it was weighted by population in the different states. In the abstract it says users of the oral contraceptive and statins were interviewed. This is missing from the study design. Were there any other limitations on who was interviewed (e.g. age)? Was it only home phone lines that were used, or mobiles as well? How many times were they called? How were the numbers selected? Who did the interviews? What were the planned numbers of participants and how was this derived? What was the nature of the previous study that fed into the questionnaire? How were responses recorded – entered into a database verbatim at the time, or digitally captured and transcribed later? Were participants paid for their time?
Is there a basis in the literature for including “neutral” and do not want to disclose” in “did not agreed”[sic]?
If you were to design a study to find out what Australian patients thought of continued dispensing of medicines for certain therapeutic areas, you would select either patients for each of the areas or a random sample of Australians. You would not use the sample that has been used here. Having said this, in the light of little other data in this area, this study probably still merits publication providing the limitations are well spelt out, and depending on the response rate.

Validity of the findings

What was the response rate for the telephone survey? How long did the telephone survey take? Demographics need to be given in more detail – assume readers will not all look at the previous paper. It would be helpful to divide these on users of oral contraceptives and users of statins. At the minimum it should state how many interviewees used oral contraceptive and how many used statins.
While interesting, getting views of two very select groups of people (those taking statins and those taking oral contraceptives) on what they think about a continued dispensing for other therapeutic areas that most have had no first-hand experience is of some concern. This select group may provide vastly different information from the general public or others with that condition. Those who have each of the conditions specified are very low in numbers. The age and gender will be quite different from the distribution of sufferers. This needs to be expanded on fully in the limitations section of the study. It is reasonable to state that while this was a very select group most of whom did not suffer from chronic diseases, given that there is little other research on continued dispensing (presumably?), it provides an indication that there could be patient support for such expansion of this initiative, but that further research in the relevant patient groups is required. There is also a need for further research on outcomes. Having patient opinions on what they may want does not necessarily align with what provides best health outcomes.
At the start of the discussion the sentence says “this study explored Australian patients’ views “. I think it should be more reflective of the respondents, noting that it was a select group. The following sentence says “previous studies” then reports a single, old study. Twenty years on we may have a very different environment to the 1980s. Is there any information on this from Australia in recent years?
Previous studies have reported that patients have difficulty in seeing their regular doctors without a prescheduled appointment.(Arber & Sawyer 1985)
The information on SABAs would be confusing to an international reader who may not know that SABAs are available as pharmacist-only medicines in Australia. I suggest you add infomraiton about this, as well as the fact that it is inhaled SABAs and that they treat asthma (for a reader who is not a health professional).
Is there other research on continued dispensing that covers patient views – if there isn’t, state that there isn’t and therefore that this is unique. If there is, include it and compare findings.
The limitations should not just refer to another paper, but specify them.
The conclusion should be more circumspect – e.g. a select group of patients seem to prefer…

Additional comments

The abstract should include the response rate.
If the response rate is particularly low, this would be of considerable concern.

---

## Round 0.2 · Minor Revisions

This has much improved, but still needs to be carefully proofread in relation to both punctuation and grammar (e.g. results, disease suffers, some very long-winded sentences).

Also, it would be good to have some more detail on the methods, specifically in relation to sampling, questionnaire development, and data handling. Could you consider using sub-sections under "Methods"?

·

Basic reporting

See general

Experimental design

See general

Validity of the findings

See general

Additional comments

As stated this is a well written paper on a new topic worthy of publication. The addition of the appendix and the extra results, expansion of methods and limitations satisfy the requested changes

Reviewer 2 ·

Basic reporting

There needs to be a final proof read, e.g.
"Their ages were disturbed almost equally into..." presumably should be distributed not disturbed.

Experimental design

The improvements are helpful.

Validity of the findings

It is noted that by telephoning people only once each and not calling back where there was no answer, an answer phone or a busy line there could have been a bias towards people who were not working, which could have affected the survey outcome. This needs to be listed as a limitation.

Additional comments

Continued dispensing occurs in at least some states in Canada (eg Ontario).

---

## Round 0.3 · accepted · Accept

Thank you very much for addressing the outstanding issues.